# Is examining children and adolescents with autism spectrum disorders a challenge?— Measurement of Stress Appraisal (SAM) in German dentists with key expertise in paediatric dentistry

**Daniela Reis**[1,2]*, **Oliver Fricke**[1,2], **Andreas G. Schulte**[3], **Peter Schmidt**[1,2,3]

1 Department of Child and Adolescent Psychiatry, Psychotherapy and Child Neurology Gemeinschaftskrankenhaus Herdecke, Herdecke, Germany, 2 Faculty of Health, Child and Adolescent Psychiatry, Witten/Herdecke University, Witten, Germany, 3 Faculty of Health, Department of Special Care Dentistry, Dental School, Witten/Herdecke University, Witten, Germany

* d.reis@gemeinschaftskrankenhaus.de

**Data Availability Statement:** Due to the strict European General Data Protection Regulation, the statement in the questionnaire to the study

## Abstract

### Objectives

This questionnaire-based validation study investigated if the dental examination of children and adolescents with autism spectrum disorder is viewed by dentists with key expertise in paediatric dentistry as a challenge or a threat in terms of transactional stress theory. The Stress Appraisal Measure (SAM) was used for this purpose and it's feasibility and validity was examined as a first part of a multi-stage process for validation in dentistry with a sample of German dentists. It has hardly been investigated how the treatment of children and adolescents with a disorder from the autism spectrum is perceived by dentists.

### Methods

An online-based survey (39 questions) plus the SAM as an add-on as well as a preceding short story of imagination on the topic (appointment for a dental check-up in a special school) were developed. Via e-mail members of the German Society of Paediatric Dentistry (DGKiZ) received a link which enabled interested members to participate. The majority of the members of the DGKiZ have additional qualifications in the treatment of children and adolescents and further training in the area of special needs care in dentistry. The data analysis was based on the SAM and its subscales.

### Results

Out of the 1.725 members of DGKiZ 92 participants (11 male, 81 female) fully completed the questionnaire and the SAM. All in all the dentists rated their own psychological and physical stress in course of treating children and adolescents with a disorder from the autism spectrum between less and partly stressful. Although the structure of the SAM could not be fully mapped by means of a factor analysis, the different ratings "challenge" or "threat" could

participants that no pseudonymised data will be passed on to third parties and the not included data sharing permissions in the participant consent, the dataset generated from this study can not be deposited in a public repository. A request for access to data for researchers who meet criteria for access to confidential data must be made to the senior author: Peter Schmidt, email: peter.schmidt@uni-wh.de, or to a representative of our Department of Special Care Dentistry, Dental School, Faculty of Health, Witten/Herdecke University, Germany (https://www.uni-wh.de/gesundheit/department-fuer-zahn-mund-und-kieferheilkunde/lehrstuehle/lehrstuhl-fuer-behindertenorientierte-zahnmedizin) and the board of the German Society of Pediatric Dentistry, Würzburg, Germany (https://www.dgkiz.de). Applicants wanting access to the dataset on which the analyses were performed must be prepared to conform to German privacy regulations. For further details, please contact e.g. the data protection officer at the Witten/Herdecke University, Germany (https://www.uni-wh.de/datenschutz/datenschutz-wiruni-whde).

**Funding:** The authors declare that the study was funded by the Department of Special Care Dentistry at Witten/Herdecke University and the Department of Child and Adolescent Psychiatry, Psychotherapy and Neurology of Childhood and Adolescence at the Gemeinschaftskrankenhaus Herdecke as part of a collaborative project between the two departments. This scientific project is financially supported by the Software-AG-Foundation based in Darmstadt/Hesse, Germany. The funders had no role in study design, data collection and analysis, decision to publish, or preparation of the manuscript.

**Competing interests:** The authors D.R. and O.F. declare no potential conflict of interests. The authors P.S. and A.G.S. declare, that they themselves are members of the surveyed dental society (DGKiZ- German Society of Paediatric Dentistry).

be comprehensibly evaluated after reading the story. The participants rated the situation from the story in general as challenging but not as threatening. Intercorrelations between the subscales of the SAM (e.g threat and centrality) of $r = .56$ showed that the scales are not clearly independent of one another. According to the transactional stress model, the SAM bases on, stress (perceived stressfulness) arises from appraisal processes (e.g. threat, controllable-by-self) that bring about a comparison between the requirements for the described situation and one's own possibilities in terms of a person-environment-fit. In the hierarchical regression a variance of $R^2 = .48$ could be explained with all six subscales (appraisal processes) to predict perceived stressfulness of the SAM within a sample of dentists.

## Conclusions

Due to the response rate the results of the SAM are not representative for all German dentists, but it offers an insight into topics of special needs dentistry in Germany that have not yet been examined. Overall, the feasibility and validity of the SAM in the context of mapping cognitive appraisal processes and stress could be confirmed. Taking into account the result that the treatment of children and adolescents with autism spectrum disorder is seen as a challenge, it is concluded that there is a need to improve the education of dental students and graduated dentists in Germany in the field of special needs dentistry.

## Introduction

In recent years, intensive efforts have been undertaken in Germany to improve dental care by and for people with disabilities [1]. In 2009 the Federal Republic of Germany ratified the Convention on the Rights of Persons with Disabilities (UN CRPD). With reference to Article 25 of the UN CRPD, the quality of medical care for people with disabilities must no longer differ from the quality of care for people without disabilities [2].

Furthermore, health services and interventions that meet the specific needs of individuals with disabilities have to be provided [2]. However, studies often indicate that people with intellectual disabilities in all age groups around the world have poorer oral health compared to the general population. This state of oral health is determined, for example, by an increased proportion of tooth loss, a poorer periodontal health and a lower proportion of restored teeth [3–5]. This finding also applies to people with intellectual disabilities from Germany [6,7]. A variety of reasons is given for this finding.

On the one hand, patient-related parameters, such as the lack of cooperation and communication, as well as inadequate hygienic ability or the need for supportive oral care, are described [8]. On the other hand, health policy and legal conditions of each state provide the framework for e.g. infrastructure, dental fees and remuneration, but also for dental teaching and university education [9]. The national and international literature described that there is an association between the professional training of dentists and how they feel when treating people with disabilities [9–13]. A study among dentists from Germany found that their subjective strain was significantly higher, the more insufficient and incomplete their own specialist knowledge on the treatment of children with disabilities was stated [13]. Alongside the treatment of children and adolescents with disabilities, treating individuals with mental health disorders and neurodevelopmental conditions such as autism spectrum disorders (ASD) [9,14–16] are mostly seen and described as a challenge [10,13,15,16]. However, there is sparse

scientific data based on whether the perception of dentists actually corresponds to a challenge from a psychological point of view. Furthermore, there are only a few studies that show which validated survey method can be used to examine stress experiences in dentistry. These studies used, for example, either objective parameters for measuring stress [17] or instruments for self-assessment [18]. Often multidimensional questionnaires are used to measure stress appraisal and stress coping [19], which contain self-descriptions of situation-specific coping thoughts or actions [20]. Examples of this are the „Ways of Coping Questionnaire"—WAYS - [21,22] or the „Coping Operations Preference Enquiry"—COPE - [23,24]. Another example is the Stress Appraisal Measure—SAM–that captures stress perceptions based on currently occurring, cognitive processing mechanisms in coping with acute stress [25,26]. These three questionnaires are based on the transactional stress theory according to Lazarus and Folkman [27,28], whereby the COPE also uses the self-regulation model of Carver and Scheier as a theoretical basis [29]. While the WAYS and COPE questionnaires ask about situational coping processes or coping strategies, the SAM focuses on a current event and has a clear subdivision of the various control options of the respondent [26]. The SAM has already been translated into German and was examined for validation [26]. In recent years the SAM has also been translated into other languages, as several international publications show [30,31]. Various international scientific author groups used the SAM, e.g. to investigate stress perceptions and stress appraisal during childbirth [31] or currently with regard to the COVID-19 pandemic [32,33].

The purpose of the present study was to check the assumption of whether a dental examination of children and adolescents with ASD is viewed by dentists as a challenge. For this purpose, the question of how physically or psychologically stressful dentists perceive the treatment of these children and adolescents was examined. Do the dentists assess this type of treatment more as a challenge or even as a threat in terms of transactional stress theory? In addition, the validity of the SAM's usefulness as a survey tool for recording and measuring stress perceptions and stress appraisal based on a dentist sample is examined.

## Materials and methods

### The questionnaire based cross-sectional survey

The presented cross-sectional study is based on a data set that was obtained as part of the online survey SoSci (SoSci Survey GmbH; Munich; Germany) among members of the German Society of Paediatric Dentistry (DGKiZ) between August and October 2020. Prior to the start of the study, a positive vote for carrying out the survey was obtained from the board of the DGKiZ. Subsequently, the invitation and the link to participate anonymously in the questionnaire based study was sent to all DGKiZ members (*n* = 1.725) via an e-mail sent by the DGKiZ board. As a result, it was not possible for the study group to draw personal conclusions about the participants in compliance with data protection regulations. Prior to answering the first question in the electronic file, the participants had to confirm that their participation was voluntary, to declare their consent and to state that they were 18 years or older. Furthermore, the board of the DGKiZ had given consensus to the publication of the data. Since the data collection was planned in accordance with the European General Data Protection Regulation and represents an expert survey, a formal application to the responsible ethics committee of the Witten/Herdecke University was not performed before the start of the project. Instead, we received written confirmation of the ethics committee of Witten/Herdecke University that there was no need for professional advice and for ethical approval in the case of anonymous surveys among employees in the healthcare sector. At this point, it is necessary to refer to a multicenter project whose results have already been published and which is methodically based on the same regulations [32].

The first part of the questionnaire was developed by the authors and comprised a total of 39 questions and was designed containing a hybrid of 5 open and 34 closed questions. In order to be able to compare the results, the development of the questionnaire was based on previous national and international studies [9,10,13,15,16]. In addition to various demographic aspects (e.g. age, gender, years of employment), the questionnaire aimed at exploring how the dental treatment of children and adolescents with various types of disabilities respectively neurodevelopmental or psychoemotional disorders is experienced by the respondents. Therefore, personal experiences and assessments, with regard to e.g. subjective burden were asked to rate on a 5-point Likert scale from 1 "not at all stressful" to 5 "very stressful".

Another focus was on mapping one's own stress perception and stress appraisal in an examination of children and adolescents with an autism spectrum disorder (ASD). The German version of the SAM was used as a survey tool for this purpose [26]. The SAM was added to the questionnaire mentioned above at the end of the online survey as an add-on. With the help of the SAM, stress was induced with a preceding short story read by the study participant. The story allows the participant to imagine an appointment for a dental check-up in a special school for children and young people with intellectual or psychoemotional development disorders. The participants were asked to imagine how they would attend this school as a dentist. The situation was described as very confusing and chaotic and ended with the first patient with an autism spectrum disorder throwing himself on the floor before the dental exam began.

After reading the short story, the respondents were asked to answer the SAM's questions. The originally English language SAM [25] is a questionnaire with 28 items, divided into seven subscales consisting of four items per scale. This measures cognitive processing mechanisms and perceived stressfulness in the event of acute stress [26]. The authors of the SAM see the transactional stress model as the basis of their questionnaire [34,35]. According to this model, stress arises from appraisal processes that bring about a comparison between the requirements for the described situation and one's own possibilities in terms of a person-environment-fit. The perception of stress varies depending on how the situation is assessed and which forms of coping are used [36]. According to the transactional stress model, a situation is perceived as irrelevant, positive or stressful in an initial appraisal. If experienced as stressful, the person is asked to distinguish whether they consider the situation to be challenging, threatening or perceived as important. In the second appraisal, the person assesses whether they have sufficient resources of their own or whether there are other options to cope with the situation. The SAM is sub-divided in the following subscales: challenge (the situation is assessed positively, in the sense that it is manageable), threat and centrality (effects and consequences of the situation) for the first appraisal of the situation. For the second appraisal, an assessment is made of one's own controllability, controllability by others and the uncontrollability of the situation (subscale 4 to 6). In addition, there is the "overall perceived stressfulness" scale, which can be viewed as a consequence of the previous appraisal of the situation. The perceived threat and centrality of a situation are seen as the most important predictors for the perception of stress [25]. This scale division was used adopted without exception in the German version [26].

## Data analysis

In two of the three studies published by Peacock and Wong, a five- or six-factor solution of the appraisal scales was given when testing the construct validity in factor analyzes. In other studies a five-factor solution was described [26,30,31]. To verify the factor structure [25], a main axis analysis with an oblique rotation (promax rotation) was carried out in the present study after analyzing the suitability of the data. In the further course of the data analysis, the items of the seven subscales of the SAM, assessed using a 5-point Likert scale from 1 "not at all" to 5

"completely", were combined into a non-weighted index by a calculated average. The seven sub-scales "threat", "centrality", "controllable-by-self", "controllable-by-others", "uncontrollable", "challenge" and "stressfulness" were examined for their scale characteristics in the present study. The Cronbach's $\alpha$ coefficient was determined to check the internal consistency of the items on a scale. According to Bühner [37] a general assessment of the internal consistency of a scale is difficult, although the information provided by Fisseni [38] for the assessment of test parameters and quality criteria can provide an orientation. According to this, values <0.80 are to be assessed as low, values between 0.80–0.90 as average and values >0.90 as high. Further-more, Pearson correlations between the seven subscales were determined. In addition, the predictability of the "stressfulness" was verified using the six appraisal scales in a hierarchical regression. The regression examined whether the "stressfulness" can be predicted using the six appraisal scales, and how high the respective predictive value of the individual predictors is. The relevant predictors were included in the regression model for predicting the"stressfulness" according to the importance of their predictive power [25,26]. The data analysis was carried out with the statistical program SPSS Version 25 (IBM SPSS Statistics 25; IBM Corporation; New York; NY; USA). When exporting data to SPSS, serial numbers are generated from the partici-pations in SoSci Survey. Duplicate entries were detected using the serial number and excluded from the analysis. Cases with missing values in the SAM subscales or in the other questions were excluded from the data analysis. An overview of the data analysis is given in Fig 1.

## Results

### Study participants

The response rate in relation to all 1.725 e-mails that had been sent out to members of the DGKiZ was thus 11.1% for all questionnaires started ($n$ = 192), and 5.3% for the fully com-pleted questionnaires. In total 92 participants (11 male; 81 female) completed the question-naire designed by the study team and as well as the SAM. The majority of the participants were between 35 and 64 years old ($n$ = 65; 71%). Furthermore, the majority of the participants ($n$ = 50; 54.3%) stated that they had more than 16 years of professional experience. Further demographic information and details of the study participants are presented in Table 1.

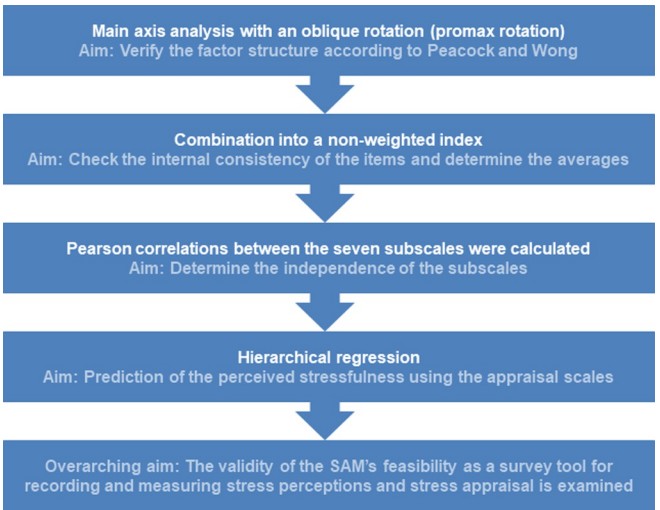

**Fig 1. Data analysis.** Notes: SAM, Stress Appraisal Measure [25], German version by [26].

**Table 1. Study participants, frequencies and percentages.**

| Study participants, n = 92 | | |
|---|---|---|
| | **n** | **Percent** |
| **Male** | 11 | 12.0% |
| **Female** | 81 | 88.0% |
| *Age (in years and in age groups)* | | |
| **under 35 (all)** | **23** | **25.0%** |
| under 35 (male) | 4 | 4.3% |
| under 35 (female) | 19 | 20.6% |
| **from 35 to 44 (all)** | **21** | **22.8%** |
| from 35 to 44 (male) | 1 | 1.1% |
| from 35 to 44 (female) | 20 | 21.7% |
| **from 45 to 54 (all)** | **36** | **39.2%** |
| from 45 to 54 (male) | 4 | 4.4% |
| From 45 to 54 (female) | 32 | 34.8% |
| **From 55 to 64 (all)** | **8** | **8.7%** |
| From 55 to 64 (male) | 1 | 1.1% |
| From 55 to 64 (female) | 7 | 7.6% |
| **65 and older (all)** | **4** | **4.3%** |
| 65 and older (male) | 1 | 1.1% |
| 65 and older (female) | 3 | 3.2% |
| *Working arrangement[a]* | | |
| Alone in his/her own practice | 39 | 42.4% |
| Employed in private practice as a dentist | 27 | 29.3% |
| Employed in a private practice as assistant dentist | 2 | 2.2% |
| Employed in a dental school at the university | 10 | 10.9% |
| Employed in a medical care center | 12 | 13.0% |
| Employed in a hospital or clinic at the university | 3 | 3.3% |
| others (e.g. students, persioner) | 4 | 4.3% |

[a] Multiple answers were possible.

## Assessment of personal psychological and physical stress

When asked how stressful the dentists rate their own psychological stress in course of treating children and adolescents with ASD, the answer options 1 "not stressful at all" to 5 "very stressful" averaged 2.74 ($SD \pm 1.06$). The assessment of one's own physical stress when treating children and adolescents with ASD yielded a mean of 2.58 ($SD \pm 1.13$). Overall, the assessments of both the psychological and the physical stress ranged between less stressful and partly.

## „Stress Appraisal Measure"(SAM)

**Factor structure and factor loadings of the SAM.** The principal axis analysis with Promax rotation was carried out with seven factors to be extracted, analogous to the SAM questionnaire. An eigenvalue >1 as a criterion for the number of factors to be interpreted was not given due the average reliability of the items. The Bartlett test for sphericity was significant ($p < .01$), indicating that the items correlate well with one another. The Kaiser-Meyer-Olkin coefficient as a measure of sample suitability was .80. Since this value was above the lower limit of .50, a factor analysis could be carried out [37]. We have to qualify that the reliability of a factor analysis depends on the sample size, but also the factor

loadings matters [39]. Guadagnoli and Velicer stated that "If a solution possesses components with only a few variables per component and low component loadings, the pattern should not be interpreted unless a sample size of 300 or more observations has been used." [40]. So we carried out the factor analysis, taking into account the factor loadings on the individual factors due to our small sample.

Due to the structure of the SAM questionnaire, seven factors were extracted that can explain 56% of the total variance before the rotation. After the rotation, the first factor alone could explain 19% and the second factor 22%, the remaining factors 15%, 15%, 16%, 8.25% and 5% of the total variance. However, since the factors correlate with one another, the total variance of all factors cannot be totaled. Table 2 illustrates the sample matrix after the factor analysis.

**Table 2. Pattern matrix principal axis factoring promax rotation–Items SAM.**

| | Factor | | | | | | |
|---|---|---|---|---|---|---|---|
| | 1 | 2 | 3 | 4 | 5 | 6 | 7 |
| SCALE 1 Item 5 feel anxious [a] | 0.61 | 0.03 | -0.02 | -0.03 | 0.40 | -0.07 | 0.07 |
| Item 11 outcome negative | 0.03 | -0.29 | 0.03 | 0.13 | 0.42 | 0.03 | -0.03 |
| Item 20 threatening situation | 0.38 | -0.01 | 0.23 | 0.22 | 0.26 | 0.22 | -0.20 |
| Item 28 negative impact | 0.03 | -0.07 | 0.02 | 0.77 | -0.10 | -0.12 | -0.31 |
| SCALE 2 Item 6 Important consequences | 0.03 | 0.00 | 0.05 | 0.75 | 0.19 | -0.01 | 0.38 |
| Item 9 Will be affected | 0.70 | -0.10 | 0.03 | 0.07 | -0.25 | -0.11 | 0.03 |
| Item 13 serious implications | 0.06 | 0.20 | -0.02 | 0.63 | 0.12 | -0.04 | -0.14 |
| Item 27 long-term consequences | 0.34 | 0.08 | -0.11 | 0.55 | -0.23 | 0.01 | 0.16 |
| SCALE 3 Item 12 Have ability to do well | -0.15 | 0.74 | 0.02 | 0.09 | -0.10 | -0.03 | -0.12 |
| Item 14 have what it takes | -0.04 | 1.09 | -0.22 | 0.06 | 0.09 | 0.10 | 0.04 |
| Item 22 Will overcome problem | 0.09 | 0.43 | 0.19 | 0.02 | -0.28 | -0.04 | 0.18 |
| Item 25 have skills necessary | -0.07 | 0.86 | -0.03 | 0.05 | 0.06 | -0.09 | 0.07 |
| SCALE 4 Item 4 someone I can turn to | -0.16 | -0.29 | 0.95 | 0.13 | -0.14 | 0.01 | 0.03 |
| Item 15 help available | -0.04 | 0.35 | 0.62 | -0.08 | 0.17 | 0.04 | 0.03 |
| Item 17 resources available | -0.19 | 0.39 | 0.27 | -0.05 | 0.14 | -0.10 | 0.04 |
| Item 23 anyone who can help | 0.06 | -0.08 | 0.71 | -0.13 | 0.04 | -0.07 | 0.27 |
| SCALE 5 Item 3 outcome uncontrollable | -0.09 | 0.12 | -0.04 | -0.01 | 0.62 | 0.03 | -0.06 |
| Item 1 totally hopeless | 0.00 | -0.15 | -0.03 | 0.04 | 0.73 | -0.09 | -0.03 |
| Item 18 beyond anyone's power | 0.02 | 0.01 | -0.01 | -0.05 | -0.03 | 0.90 | 0.03 |
| Item 21 problem unresolvable | -0.26 | -0.09 | -0.00 | 0.50 | 0.04 | 0.33 | 0.04 |
| SCALE 6 Item 7 positive impact | -0.10 | -0.01 | 0.17 | -0.15 | -0.06 | 0.08 | 0.47 |
| Item 8 Eager to tackle | 0.30 | 0.29 | 0.39 | -0.05 | -0.25 | 0.05 | -0.06 |
| Item 10 Can become stronger | 0.16 | 0.17 | 0.10 | 0.11 | -0.11 | -0.04 | 0.39 |
| Item 19 Excited about outcome | 0.85 | 0.05 | 0.08 | -0.04 | 0.01 | 0.15 | 0.00 |
| SCALE 7 Item 2 tension caused by the situation | 0.56 | -0.18 | -0.06 | -0.11 | 0.25 | -0.13 | -0.02 |
| Item 16 resources put to the test | 0.40 | -0.08 | -0.20 | -0.11 | -0.03 | 0.23 | 0.10 |
| Item 24 stressful situation | 0.55 | -0.03 | -0.09 | 0.14 | 0.00 | -0.02 | -0.18 |
| Item 26 Efforts to cope with | 0.32 | -0.05 | -0.10 | 0.08 | -0.12 | -0.07 | 0.07 |

Notes: threat = scale 1; centrality = scale 2; controllable-by-self = scale 3; controllable-by-others = scale 4; uncontrollable = scale 5; challenge = scale 6; perceived stressfulness = scale 7, SAM, Stress Appraisal Measure, [25]., German version by [26].

Factor extraction: Principal Axis Factor Analysis.

Method of factor rotation: Promax with kaiser normalization.

a. The rotation has converged in 8 iterations.

Table 2 shows the following results:

1. Subscale "threat" scale 1: Two of the four items in the "threat" subscale have the highest load on factor 1, item 11 has the highest load on factor 5 and item 28 has the highest load on the fourth factor.

2. Subscale „centrality"scale 2: Three of the four items load the highest at factor 4, item 9 load the highest at factor 1.

3. Subscale „controllable-by-self"scale 3: All four items load the highest on the second factor.

4. Subscale „controllable-by-others"scale 4: Three of the four items load the highest on factor 3, item 17 load on the second factor the highest.

5. Subscale „uncontrollable-by-anyone"scale 5: Two items load the highest on the 5th factor, item 18 on the 6th factor, item 21 on the 4th factor.

6. Subscale „challenge"scale 6: Two items load the highest on the 7th factor, item 8 on the 3rd factor, item 19 on the first factor the highest.

7. Subscale „overall perceived stressfulness"scale 7: All 4 items load the highest on the 1st factor.

**Psychometric properties of the SAM subscales.** The reliability of the subscales, measured using the Cronbach's $\alpha$, was shown as follows: "threat": $\alpha$: .69, "centrality": $\alpha$: .76, "controllable-by-self ": $\alpha$: .89, "controllable-by-others ": $\alpha$: .82, "uncontrollable-by-anyone" $\alpha$: .56. The subscale "challenge" had insufficient reliability with all four items ($\alpha$: .33), especially item 19 "excited about outcome" correlated negatively with two items on its own scale. The subscale "overall perceived stressfulness" had an internal consistency of $\alpha$: .66.

**Intercorrelations of the SAM subscales.** The subscales formed [25] do not show a relative independence of the individual subscales from one another in all cases according to the available intercorrelations (Table 3). The "threat" subscale, for example, has a strongly positive correlation with the "centrality" and "uncontrollable" of the situation. In addition, the subscale "controllable-by-self" has a highly positive correlation with "controllable-by-others".

As expected, the "overall perceived stressfulness" correlates highly positively with the subscales "threat", "centrality" and "uncontrollable" and highly negative with "controllable-by-self" and "controllable-by-others". The assessment of the situation as challenging is not related to the assessment of the situation as threatening ($r$ = -.01).

**Table 3. Intercorrelations of the SAM subscales.**

|  | 1 | 2 | 3 | 4 | 5 | 6 |
|---|---|---|---|---|---|---|
| **1 threat** |  |  |  |  |  |  |
| **2 centrality** | .56 |  |  |  |  |  |
| **3 controllable-by-self** | -.62 | -.23 |  |  |  |  |
| **4 controllable-by-others** | -.43 | -.28 | .51 |  |  |  |
| **5 uncontrollable** | .50 | .27 | -.51 | -.36 |  |  |
| **6 challenge** | -.01 | .21 | .31 | .36 | -.14 |  |
| **7 perceived stressfulness** | .63 | .48 | -.51 | -.43 | .32 | .01 |

Notes: n = 92.

SAM = Stress Appraisal Measure, (5-point Likert scale from 1 "not at all" to 5 "completely"), [25], German version by [26].

**Prediction of the overall perceived stressfulness.** For the hierarchical regression to predict the "overall perceived stressfulness" by the six other subscales, several preconditions were checked. The precondition, that the residuals are normally distributed [39], was checked using the Q-Q plot. The result of the residual test of the dependent variable "overall perceived stressfulness" showed no deviation from the normal distribution. The Shapiro-Wilk test as a test for normal distribution of the residuals indicates that the distribution of the scores is not different from a normal distribution ($p = .17$). The homoscedasticity, defined as the independence of the scatter of the measurement errors, was checked via the Breusch-Pagan test. The null hypothesis of this test is that there is homoscedasticity ($p = .26$). Multicollinearity describes the correlation between the predictors being so high that the estimation of the individual coefficients is deemed inaccurate [39]. A test function for multicollinearity is the Variance Inflation Factor (VIF). Values greater than 10 are considered problematic [41]. The values of the VIF were all well below 10 indicating no multicollinearity.

To calculate hierarchical multiple regression, variables were included according to the importance of their predictive power. When predicting "stressfulness", the "threat" subscale (in the first model) and the "centrality" subscale (in the second model) were included. In the third model, all four other subscales were recorded. It was found: a) in the first model, a high degree of variance explanation of the „stressfulness" through the perception of the threat of the situation ($\beta = .64$, $R^2 = .41$, $p < .01$). b) By adding the "centrality" subscale in a second model, the influence of the new predictor ($p < .10$) on the $\beta$-weight of "threat" was confirmed. The $\beta$-weight of the "threat" decreased slightly ($\beta = .54$ instead of .64) and both subscales had an explanation of variance of $R^2 = .43$. c) When adding the remaining four subscales in a third model, a negative influence of the predictors "controllable-by-self" ($\beta = -.21$, $p < .10$) and a non-significant negative effect of the subscale "controllable-by-others" ($\beta = -.17$) can be confirmed. The subscales "uncontrollable-by-anyone" and "challenge" did not explain any significant variance. The total explained variance of the third model shows an $R^2$ of .48.

## The short story

The mean values of the seven SAM subscales given by the participants after having read the short story are shown in Table 4.

The participants rated the given situation on average less as threatening but more as challenging. They also found the scenario to be less significant for them in terms of its consequences and effects. In addition, they were more positive about coping with the problem due to their own skills and possibilities. The same applied to the perspective of whether there were enough resources and skills available from the other side to cope with the situation.

Table 4. SAM subscales-mean values and standard deviations.

| Scale | Mean value | Standard deviation |
|---|---|---|
| threat | 1.47 | 0.55 |
| centrality | 1.68 | 0.70 |
| controllable-by-self | 3.82 | 0.84 |
| controllable-by-others | 3.38 | 0.97 |
| uncontrollable | 1.75 | 0.68 |
| challenge | 2.97 | 0.63 |
| perceived stressfulness | 2.68 | 0.76 |

Notes: n = 92.

SAM = Stress Appraisal Measure, [25], German version by [26].

## Discussion

### Assessment of personal psychological and physical stress

When the dentists with key expertise in paediatric dentistry were asked in our study how they assessed their psychological and physical stress in course of treating children and adolescents with autism spectrum disorders (ASD), they indicated to perceive a less stressful or moderate psychological stress. This was also true in regard to physical stress. Previous studies dealt with the question of how challenging the treatment of children and adolescents with ASD is for dentists. These were either concrete challenges such as the behavior of children [42] or the need for information or further training on the subject or practical recommendations for action [11,14,15].

Studies about psychological and physical stress and the resulting stress experience of dentists when treating children and adolescents without disabilities showed that there is an association between the stress or stress experience and the practical experience of dentists, the procedures used and the age of the patient [43–45]. In terms of practical experience, the emphasis here is primarily on expertise in dealing with patients with ASD [46]. According to an US study, one suggestion for improving knowledge and practical experience in this area would be interdisciplinary cooperation with professions such as occupational therapy or psychology [47]. Since the majority of our study participants already had a professional experience of more than 16 years (n = 50, 54.3%), practical experience in dealing with patients with ASD may also have been a decisive factor here for the rather low exposure values. It should also be noted that the majority of respondents were female dentists. Most of the participants (men and women) were between 45 and 54 years old. The age and therefore the associated professional experience, as noted above [43], and possibly also gender [45] can have an influence on personal psychological and physical stress.

### „Stress Appraisal Measure"(SAM)

The internal consistencies of six of the seven scales were between 0.56 and 0.89. According to Fisseni, the subscales "controllable-by-self" and "controllable-by-others" showed an average reliability [38]. The other subscales showed low reliability. The subscale "uncontrollable", comparable to studies 1 and 3 [25], only has an internal consistency of 0.56. The "challenge" scale in Delahaye et al. had an $\alpha$ value of 0.57 [26]. In our study, the internal consistency is even lower ($\alpha$: 0.33). Mainly because of the negative correlations with two items on our own scale, item 19 could have been excluded from the further analyzes. However, since the full SAM tool was to be investigated, this possibility was abandoned. Not all of the Peacock and Wong factors could be replicated with the principal axis analysis [25]. An examination of the factorial validity and dimensionality of the SAM favors, for example, a 4-factor solution of the appraisal scales and criticizes the partly redundant factors as well as the low internal consistency of the factors in the original study [48]. The "overall perceived stressfulness" (factor 1) and "controllable-by-self" (factor 2) could be replicated in our study with the respective four items. Item 17 (controllable-by-others) "resources available" loaded the highest on the second factor. Perhaps the wording of the item was unclear, so that the majority of participants related it to themselves rather than to other resources. Item 9 (centrality) "will be affected" loaded the highest on factor 1, which indicates that the importance of a potentially challenging situation is closely related to the experience of stress. The same applies to two items on the "threat" scale; which also loaded the highest on the first factor. Item 19 (challenge) had the highest load on the first factor, "overall perceived stressfulness". Since excitement goes hand in hand with the feeling of stress, this high charge is understandable. Overall, however, the various assessments and the general

experience of stress could be well mapped separately from one another by the scenario, which suggests that the content of the questionnaire with its subscales is well applicable. Although the results do not speak for a relative independence of the individual subscales in all cases, a variance of $R^2$ = .48 could be explained in the hierarchical regression with all six subscales. Similar to the results of previous studies, the predictors "threat" and "centrality" were the relevant predictors for experiencing stress [25,26,31]. The appraisal of "controllable-by-self" had a negative effect on the experience of stress, at least in the marginally significant range.

## The short story

The described situation in the short story was seen as challenging by the participants (mean: 2.97). Furthermore, this was perceived as not particularly threatening (mean: 1.47). The information matches the responses of the participants on the perception of their own psychological and physical stress when treating children and adolescents with ASD. The results of the present study, which the SAM provides as an evaluation tool, can thus confirm for the first time the assumption that a dental examination of children and adolescents with ASD by dentists is a challenge. It was already shown in other studies (e.g. on birth) that the SAM can validly map sensations. A Portuguese study described that expectant parents perceive the birth of a child mainly as a challenge and only rarely as a threat [31]. One reason why a situation is perceived more as challenging and less as threatening could be that the interviewed person is able to cope with the situation in a problem-oriented manner, e.g. by making plans to solve the stressful experience and thus accept the challenge. Emotion-oriented coping, on the other hand, rather includes the regulation of the negative emotions caused by the situation and is related to the perception of the threat of the situation and the experience of stress [26]. The missing correlations in our study between "threat" and "challenge" make it clear that these perceptions are not related to one another. In a focus group, dentists and the dental team reported that it makes sense and is an investment in the future to take time for these patients. According to the study, the experience with dental treatments of persons with ASD can be improved by good preparation for the appointments and by education about ASD [49]. Five overarching issues are identified by dentists as the challenges in treating patients with ASD: 1.) each patient with ASD has their own needs, 2.) communication plays a key role, 3.) specific techniques for ASD are important, 4.) a conflict between needs and ressources and finally 5.) the personal reward for the work [50].

## The study

As a limitation of the present study, it should be mentioned that this study was only carried out among members of DGKiZ which represents only one out of several dental associations for German dentists. Mainly, persons with key interest and key expertise in paediatric dentistry join this association. These dentists are generally very experienced in dealing with children and adolescents, including children with underlying diseases. It is very pleasant for the author team that the group of German dentists with key expertise in paediatric dentistry confirm the expected result of good stress resistance in dental care of children and adolescents with ASD. If validation had not been successful in this group, further validation in larger and other groups of dentists from Germany would probably not have been necessary. Now, in order to continue the process of validating the SAM in dentistry and to broaden the topic overall, a survey has already been carried out for German dentists from the public health service. A survey including the SAM for general German dentists is being planned. Moreover surveys about the dental care situation have also already been carried out, with the concerned parents and caregivers of e.g. persons with ASD. In addition, primary data collection on dental care and prevention in

children, adolescents and younger adults with neurodevelopmental or intellectual disabilities (in particular with ASD) in Germany is planned. Furthermore, it should be discussed whether these planned surveys should be carried out online-based or in paper-pencil style, since a response rate of the participating members of 5.3% percent with a fully completed questionnaire can be viewed as below average. We only recruited the participants online. More than 190 persons started to complete the online questionnaire. Due to a high number of missing values, many participants could not be included in this analysis. Therefore, the SAM results of the present study cannot be regarded as representative for all German dentists but it offers an insight into important topics in relation to special needs dentistry in Germany that have not yet been examined. In order to obtain a higher participation rate for further projects it might be useful to recruit participants not only via e-mail but also at conferences.

## Conclusion

Cognition and processing of stress can be measured feasibly and with sufficient validity using "Stress Appraisal Measure" (SAM) also in dentists. The underlying model of stress response is reconfirmed for SAM. Factor analyses and the SAM analysis reveal that dental diagnostic procedures in children and adolescents with ASD are perceived more as challenging than as threatening situation by German dentists with key expertise in paediatric dentistry. Therefore, special training sessions in special needs dentistry are recommended for all dentists who are involved in the treatment of children and adolescents with ASD.

## Acknowledgments

The authors are grateful to all members, the office employees in Würzburg, Germany and the board of the German Society of Paediatric Dentistry for their cooperation, constribution and finally realization of this study. We are also grateful for the linguistic support from Kurt Mathisen and the support with SoSci Survey from Marie-Lené Scheiderer.

## Author Contributions

**Conceptualization:** Oliver Fricke, Andreas G. Schulte, Peter Schmidt.

**Data curation:** Daniela Reis, Peter Schmidt.

**Formal analysis:** Daniela Reis, Peter Schmidt.

**Funding acquisition:** Oliver Fricke, Andreas G. Schulte.

**Investigation:** Daniela Reis, Peter Schmidt.

**Methodology:** Daniela Reis, Oliver Fricke, Andreas G. Schulte, Peter Schmidt.

**Project administration:** Peter Schmidt.

**Resources:** Oliver Fricke, Andreas G. Schulte, Peter Schmidt.

**Software:** Daniela Reis.

**Supervision:** Oliver Fricke, Andreas G. Schulte, Peter Schmidt.

**Validation:** Daniela Reis.

**Visualization:** Daniela Reis.

**Writing – original draft:** Daniela Reis.

**Writing – review & editing:** Oliver Fricke, Andreas G. Schulte, Peter Schmidt.

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
