## [Decision Letter · Decision Letter 0]

9 Mar 2022

PONE-D-21-39587Measurement of Stress Appraisal (SAM) of German Dentists Examining Children and Adolescents with Autism Spectrum DisordersPLOS ONE

Dear Dr. Reis,

Thank you for submitting your manuscript to PLOS ONE. After careful consideration, we feel that it has merit but does not fully meet PLOS ONE’s publication criteria as it currently stands. Therefore, we invite you to submit a revised version of the manuscript that addresses the points raised during the review process.

Having intensively double checked your re-submitted draft, two of our external reviewers have forwarded strongly contrasting recommendations. Consequently, I also have inspected your submission (see R #1), to come to a more balanced decision. Please note that your manuscript would not seem satisfying, and is not considered ready to proceed. Indeed, some most critical aspects would seem in need of a thorough discussion. With your re-revision, you should follow the reviewers' comments added below, to finalize your paper convincingly, and to meet both Plos One's quality standards and our readership's expectations.

In order to expedite the processing of the revised manuscript, please make sure to address each of the criticized aspects and incorporate all your carefully elaborated responses within the manuscript. Your rebuttal letter should include point-by-point responses to the reviewer comments, even if you happen to disagree with them, or feel not being able to incorporate all the suggested feedback. Thus, I would like to encourage you to provide a thorough (in terms of language, reviewers' constructive criticism, content, generalizable outcome, and/or Authors' Guidelines) revision in order to avoid an iterative and lengthy review process and facilitate a smooth publication process. Please remember that Plos One will not provide an in-depth copy-editing service, and this requires flawless manuscripts to proceed.

We look forward to receiving your revised manuscript.

Kind regards,

Andrej M Kielbassa, Prof. Dr. med. dent. Dr. h. c.

Academic Editor

PLOS ONE

Journal Requirements:

a) Did participants provide their written or verbal informed consent to participate in this study?

 “The authors declare that the study was funded by the Department of Special Care Dentistry at Witten/Herdecke University and the Department of Child and Adolescent Psychiatry, Psychotherapy and Neurology of Childhood and Adolescence at the Gemeinschaftskrankenhaus

Herdecke as part of a collaborative project between the two departments. This scientific project is financially supported by the Software-AG-Foundation based in Darmstadt/Hesse, Germany.”

5. PLOS requires an ORCID iD for the corresponding author in Editorial Manager on papers submitted after December 6th, 2016. Please ensure that you have an ORCID iD and that it is validated in Editorial Manager. To do this, go to ‘Update my Information’ (in the upper left-hand corner of the main menu), and click on the Fetch/Validate link next to the ORCID field. This will take you to the ORCID site and allow you to create a new iD or authenticate a pre-existing iD in Editorial Manager. Please see the following video for instructions on linking an ORCID iD to your Editorial Manager account: https://www.youtube.com/watch?v=_xcclfuvtxQ.

Reviewers' comments:

Reviewer's Responses to Questions

**Comments to the Author**

1. Is the manuscript technically sound, and do the data support the conclusions?

Reviewer #1: No

Reviewer #2: No

Reviewer #3: Yes

2. Has the statistical analysis been performed appropriately and rigorously? 

Reviewer #1: Yes

Reviewer #2: I Don't Know

Reviewer #3: I Don't Know

3. Have the authors made all data underlying the findings in their manuscript fully available?

Reviewer #1: No

Reviewer #2: No

Reviewer #3: Yes

4. Is the manuscript presented in an intelligible fashion and written in standard English?

Reviewer #1: Yes

Reviewer #2: No

Reviewer #3: Yes

5. Review Comments to the Author

Reviewer #1: Abstract

- "(...) sent to members of the German Society of Paediatric Dentistry (DGKiZ) via a link to participate." Please add number of members having been invited to participate.

- "92 participants (11 male, 81 female) fully completed the questionnaire." Must read "Ninety-two participants (...)", please re-edit. Indeed, 92 would seem a poor number only. Why do you think that this would result in some kind of representative outcome?

- "(...) and in some cases there were high intercorrelations between the scales, (...)." Please provide exact results, give r and p values. Phrases like "some cases" or "high intercorrelations" would not seem acceptable. Please remember that future readers will decide switching to your full text AFTER having read your Abstract section.

- "This could be reasoned in the circumstance that the participants in the present study were generally very experienced in treating children, including children e.g. with underlying diseases." This would not seem astonishing. Do you see any news from your study, not only confirmative ones?

- "In future, dentists who rarely treat children could also be surveyed in a further, larger study." Do not stick to meaningless phrases here. With your Conclusions, please stick exclusively to your revised aims (see comments given below). Do not simply repeat your results (or even speculate) here. Instead, provide a reasonable and generalizable extension of your outcome.

Intro

- This study has been finished, right? "The purpose of the present study is to check (...)" must read "The purpose of the present study was to check (...)".

- Same with "(...) is examined."

- Both aims and objectives elaborated here would seem convincing. However, "SAM’s utility" aspects should be mentioned with your Abstract section, too. Please revise/adapt carefully.

Meths

- Do not use legal terms like GmbH, ®, and so on.

- With ALL materials and methodologies (including statistical software), please use general names with your text, followed by (brand name; manufacturer, city, ST [if US], country) in parentheses. Stick to semicolon. Revise thoroughly.

- "Before the start of the study, a positive vote for carrying out the survey was obtained from the board of the DGKiZ." What about the positive vote of an Ethical Committee? Please provide vote number and date of approval.

- "Since the data collection was planned in accordance with the European General Data Protection Regulation and represents an expert survey, a formal ethics application to the responsible ethics committee of the Witten/Herdecke University was not performed before the start of the project." This would seem astonishing. An "expert survey" does not need an ethical approval? Please provide a sound and reliable reference.

- "(...) successfully sent to 1.725 e-mail addresses (...)." Please see comments given above. With 92 respondents, some 5% have participated only, and this would not seem a reliable database, don't you think so?

- "(...) This measures cognitive processing mechanisms and perceived stressfulness in the event of acute stress [26]. Peacock and Wong see the transactional stress model as the basis of their questionnaire [34,35]. According to this model, stress arises from (...)." This does not seem to refer to your methodology, but might be discussed later. Please revise carefully, and focus on the methodology here.

- Reduce repeated mentioning of Author names, please (see, for example, "Peacock and Wong", but this also refers to other names). Instead, please focus on your main thoughts, to ensure readability (Authos' previous work will be acknowledged with your References section).

- Legend Fig 1: Please do not repeat the full reference here, see comments given above. [Number] will be sufficient.

- Same with legend of Table 2 and Table 4.

Results

- Revise carefully for any typos. See, for example, Table 1, "(e.g. students. persioner)".

- "The Kaiser-Meyer-Olkin coefficient as a measure of sample suitability was .80. Since this value was above the lower limit of .50, a factor analysis could be carried out [37]." Please provide more information on the "suitability". Again, see comments given above, and remember that 92 respondents would not seem convincing.

- Please double check, and revise for correct inter punctuation. See "(β = .64. R2 = .41 p <.01)".

- Again, several aspects should be explained and discussed in the Discussion section. Here, please focus on the results.

Disc

- "To our best knowledge, there is no study that specifically deals with the psychological and physical stress on dentists from Germany when treating children and adolescents with ASD." It would seem unclear why "dentistry from Germany" have been considered important for the Authors. What would predispose "German dentists" from other nations? What about the transferability of your outcome to other dentists from other countries?

- "This study now offers this information for the first time." And now? Again, why do you think that his would be important?

- Again, please revise for minor shortcomings, see "Since the majority of our study participants already had of professional experience of (...)."

Concl

- Remember that this section is not a second summary. Again, with your Conclusions, please stick exclusively to your revised aims. Do not simply repeat your results (or even speculate) here. Instead, provide a reasonable and generalizable extension of your outcome, which must be based on your results.

- The current version of this section would seem right, but major aspects must be provided with the Discussion section, since these are not considered conclusions from your study.

Refs

- Please stick to the Journal guidelines, and consults some recently published Plos One papers.

- Style would be "Tanagawa M, Yoshida K, Matsumoto S, Yamada T, Atsuta M. Inhibitory effect of antibacterial resin composite against Streptococcus mutans. Caries Res. 1999; 33(5): 366–371. https://doi.org/10.1159/ 000016535 PMID: 10460960" Revise thoroughly.

In total, this submitted draft would seem interesting, is considered easily intelligible, and should be worth following after a thorough revision, following the aspects given above. Additionally, the manuscript is ready for external review.

Reviewer #2: Many thanks for asking me to review this paper.

Overall summary

While I acknowledge there is little data available on this topic in Germany, overall I do not feel this paper merits publication in a prestigious journal like PLOS ONE. The questionnaire response rate of 5% is inadequate and liable to survey error and a high degree of reporting bias.

It may have been more prudent for the investigators to find new ways of engaging with the survey population to yield an improved response rate to the survey, rather than attempt to publish a questionnaire survey with such a poor response rate. As almost 95% of the survey population did not respond, I do not feel any credible conclusions can be drawn on the views of German paediatric dentists towards treating children and adolescents with autism spectrum disorders.

I have some further more specific comments to share below that may help to improve the paper, should attempts be made to improve its methodological rigour:

The title of the paper could be revised to represent the fact that it is a questionnaire study and restricted to paediatric dentists.

The paper is difficult to follow and would benefit from a technical restructure to provide a clearer rationale for the research question, clearer explanation of background context and greater clarity over terms used to define the population of interest and why this particular methodology was chosen. There may be more appropriate methodologies for investigating this topic and it doesn’t feel like this approach has elucidated the views of German paediatric dentists in a complete and comprehensive way. The methodology appears overly complex for such predictable results, which leads me to think why this research was necessary.

I fully appreciate the authors are using a second language, but in places it is not clear what is meant, in particular in the background are references being made specifically in regard to autism spectrum disorders or disability/mental disorders more generally?

In terms of the methods there are insufficient details of the questionnaire development (piloting and refinement of the tool), administration (for example how were duplicate entries managed, or ineligible individuals accounted for), anonymity, storage and management of the data, and there is no reference to efforts to maximise the response rate. A recruitment diagram would aid reader understanding of the process and response and completion rates. Particularly as the response rate is only referred to in the conclusion section. In addition, a copy of the questionnaire should be provided as an online appendix to accompany the paper.

The rigour of the results and corresponding conclusions can only be appraised based on a more robust response rate, which is not observed in this work.

Reviewer #3: Summary

The manuscript is well written and particularly scientifically sound as the results of the study are given in elaborate detail. The data analysis is very extensive and the regression analysis is discussed in great depth. The study clearly highlights the novel element of using the SAM as a survey tool for measuring stress perceptions and stress appraisal.

Minor Revision

However there are few points that still need further elaboration:

1. It would be helpful for the reader if the authors added a few lines explaining the terms used for the seven subscales (threat, centrality, controllable-by-self, controllable-by-others, uncontrollable, challenge and stressfulness) and what they denote in light of the study.

2. There is a very detailed representation of data in Table 1 regarding the demographic aspects of the study but no mention of these results in the discussion. It would be interesting to see how age, gender and the working arrangement of the dentists relate to the psychological and physical stress experienced by them when treating children with ASD. There is a clear majority of females in the study and its relation to the results must be highlighted in the discussion.

3. More references of studies done in other parts of the world can strengthen the discussion especially those studies where dentists having less experience are questioned for the study.

4. Line 65, the word should be ‘based’

5. Line 113, what is the meaning of ‘resp’ or is it a typo?

6. Line 338, a in place ‘of’

7. Line 393, ‘states’ instead of ‘statements’.

6. PLOS authors have the option to publish the peer review history of their article (what does this mean?). If published, this will include your full peer review and any attached files.

Reviewer #1: No

Reviewer #2: No

Reviewer #3: **Yes: **Dr Fatima Suhaib

---

## [Author Response · Author response to Decision Letter 0]

22 Apr 2022

We respond to specific reviewer and editor comments in the "answers to the reviewer"

---

## [Decision Letter · Decision Letter 1]

26 May 2022

PONE-D-21-39587R1Is examining children and adolescents with autism spectrum disorders a challenge? - Measurement of Stress Appraisal (SAM) in German dentists with key expertise in paediatric dentistryPLOS ONE

Dear Dr. Reis,

Thank you for submitting your manuscript to PLOS ONE. After careful consideration, we feel that it has merit but does not fully meet PLOS ONE’s publication criteria as it currently stands. Therefore, we invite you to submit a revised version of the manuscript that addresses the points raised during the review process.

I have read your re-submitted version, to double check your revisions prior to forwarding your paper to the reviewers (see R #1). Having intensively reviewed your revised draft, our external reviewers differed considerably with their final recommendations. Please note that your current version still would benefit from thorough re-edits, please see the comments of one of external reviewers below. Thus, I would like to encourage you to provide a thorough (in terms of language, reviewers' constructive criticism, content, generalizable outcome, and/or Authors' Guidelines) revision in order to avoid an iterative and lengthy review process and facilitate a smooth publication process. Please note that a further non-conning version of your draft must lead to outright reject. Comments from external reviewer:While I acknowledge the authors have made significant attempts to address my points and there are improvements to the manuscript, I am still of the opinion that the study lacks meaningful results due to coverage issues which have limited impact. 

Although the response rate to the survey is mentioned more explicitly as a key limitation in the study, and it is now mentioned appropriately in the abstract, reference to it is still missing from the results section. An overall response rate should be presented in the first line of the results section to ensure readers understand the context to the results and how many respondents they relate to, this is a key expectation in cross sectional studies. I am aware that many journals will not publish a cross sectional survey with such a low response rate and I am not convinced the authors responses adequately address the issue of quality in their approach. 

The results of the study are not particularly remarkable, as one would expect dentists with an interest in paediatric dentistry to experience less stress when treating children and adolescents with ASD. Again, this for me is a design flaw and the study might have applied an alternative design to elucidate the views of different type of dentists, which might of provided some greater insight into the issue. Indeed, the conclusion of the paper appears to recommend 

“special training sessions in special needs dentistry for dentists involved in the treatment of children and adolescents with ASD” 

but I am not sure the results of this study substantiate this point and it is not clear if the paper is recommending more training for paediatric dentists or the general dental population. I suspect it is the latter but that cannot be made based on the results of this study. This study provides only data on the utility of SAM in measuring stress response and the perceived levels of stress in a small sample of pediatric dentists.

We look forward to receiving your revised manuscript.

Kind regards,

Andrej M Kielbassa

Academic Editor

PLOS ONE

Reviewers' comments:

Reviewer's Responses to Questions

**Comments to the Author**

1. If the authors have adequately addressed your comments raised in a previous round of review and you feel that this manuscript is now acceptable for publication, you may indicate that here to bypass the “Comments to the Author” section, enter your conflict of interest statement in the “Confidential to Editor” section, and submit your "Accept" recommendation.

Reviewer #1: All comments have been addressed

Reviewer #2: All comments have been addressed

Reviewer #3: (No Response)

2. Is the manuscript technically sound, and do the data support the conclusions?

Reviewer #1: Yes

Reviewer #2: Partly

Reviewer #3: Yes

3. Has the statistical analysis been performed appropriately and rigorously? 

Reviewer #1: Yes

Reviewer #2: I Don't Know

Reviewer #3: I Don't Know

4. Have the authors made all data underlying the findings in their manuscript fully available?

Reviewer #1: Yes

Reviewer #2: Yes

Reviewer #3: No

5. Is the manuscript presented in an intelligible fashion and written in standard English?

Reviewer #1: Yes

Reviewer #2: Yes

Reviewer #3: Yes

6. Review Comments to the Author

Reviewer #1: With the help of the reviewers, this revised and re-submitted draft has been considerably improved, and is considered ready for external review.

Reviewer #2: (No Response)

Reviewer #3: While the authors have made some improvemnets in the article however I still feel that there should be mention of similar studies done in other parts of the world as it gives a bigger picture of the challenges faced in dealing with children with Autism.

If the the authors can highlight this point then it is acceptable for publication.

7. PLOS authors have the option to publish the peer review history of their article (what does this mean?). If published, this will include your full peer review and any attached files.

Reviewer #1: No

Reviewer #2: No

Reviewer #3: No

---

## [Author Response · Author response to Decision Letter 1]

23 Jun 2022

Dear Professor Kielbassa, dear reviewers,

we would like to thank you for the helpful notes and comments on our submitted manuscript. We hope that we were able to meet the editorial requirements of the journal as well as the needs of the reviewers to improve the manuscript in our revision. 

Comments from external reviewer:

While I acknowledge the authors have made significant attempts to address my points and there are improvements to the manuscript, I am still of the opinion that the study lacks meaningful results due to coverage issues which have limited impact. 

Thank you very much for your review. Thank you for acknowledging our efforts to improve the manuscript based on your comments. With regard to the sample size and the generalization of the results to the population of German dentists with key interest and key expertise in paediatric dentistry or to general dentists, the impact of the results can certainly be regarded as limited. However since the question of stress vs. challenge has not yet been scientifically investigated, but has always been spoken of as a challenge based on experience, we assume that our manuscript has a scientific impact. 

In lines 441 to 451 we address the fact that further research work of our scientific group will advance the validation process of the SAM in dentistry. Furthermore, we are working on the topic of examination of patients with ASD in dentistry in further surveys (441-451, page 22). 

The results of the feasibility of the SAM are meaningful in our opinion, as they confirm similar results from other studies on the feasibility with a different sample. Cognition and processing of stress can be measured feasibly and with sufficient validity using the "Stress Appraisal Measure" (SAM) also in dentists. 

Finally, we would like to point out that in the few studies on the subject of "attitudes of dentists to the treatment of children with ASD" also small sample sizes were reported. For example the study by Weil et al 2010 „Treating patients with autism spectrum disorder -SCDA members' attitudes and behavior“ only reported a sample size of 75 members of the society .

Although the response rate to the survey is mentioned more explicitly as a key limitation in the study, and it is now mentioned appropriately in the abstract, reference to it is still missing from the results section. An overall response rate should be presented in the first line of the results section to ensure readers understand the context to the results and how many respondents they relate to, this is a key expectation in cross sectional studies. I am aware that many journals will not publish a cross sectional survey with such a low response rate and I am not convinced the authors responses adequately address the issue of quality in their approach. 

We presented the overall response rate (for the fully completed questionnaires and for all questionnaires started) in the first line of the results section (219-221, page 10). 

The results of the study are not particularly remarkable, as one would expect dentists with an interest in paediatric dentistry to experience less stress when treating children and adolescents with ASD. Again, this for me is a design flaw and the study might have applied an alternative design to elucidate the views of different type of dentists, which might of provided some greater insight into the issue. Indeed, the conclusion of the paper appears to recommend “special training sessions in special needs dentistry for dentists involved in the treatment of children and adolescents with ASD” but I am not sure the results of this study substantiate this point and it is not clear if the paper is recommending more training for paediatric dentists or the general dental population. I suspect it is the latter but that cannot be made based on the results of this study. This study provides only data on the utility of SAM in measuring stress response and the perceived levels of stress in a small sample of pediatric dentists.

For incremental validation, we started with a small group of dentists (those with expertise in paediatric dentistry). If validation had not been successful in this group, further validation in larger and other groups of dentists from Germany would probably not have been necessary.

Now, our scientific working group is planning further surveys on this subject and same questions with general dentists and has already carried out surveys with dentists in the public health sector and also with people working in child and adolescent psychiatry, psychosomatic medicine and psychotherapy in Germany. 

Our assumption is that general dentists experience a higher level of stress when treating children and adolescents with ASD than dentists with key expertise in paediatric dentistry. This must now be checked in one of the next steps. Nevertheless, we recommend, in the knowledge from the professional (clinic and research everyday life), that further training on this topic are recommended for both groups (general dentists and dentist with key expertise in paediatric dentistry) or for all dentists.

6. Review Comments to the Author

Reviewer #1: With the help of the reviewers, this revised and re-submitted draft has been considerably improved, and is considered ready for external review.

Thank you very much for your review. 

Reviewer #2: (No Response)

Reviewer #3: While the authors have made some improvements in the article however I still feel that there should be mention of similar studies done in other parts of the world as it gives a bigger picture of the challenges faced in dealing with children with Autism. If the the authors can highlight this point then it is acceptable for publication.

Thank you very much for your review. We researched further studies on the topic and included them in the manuscript.

---

## [Decision Letter · Decision Letter 2]

30 Jun 2022

Is examining children and adolescents with autism spectrum disorders a challenge? - Measurement of Stress Appraisal (SAM) in German dentists with key expertise in paediatric dentistry

PONE-D-21-39587R2

Dear Dr. Reis,

We’re pleased to inform you that your manuscript has been judged scientifically suitable for publication and will be formally accepted for publication once it meets all outstanding technical requirements.

Congratulations, and stay healthy, please...

Prof. Dr. med. dent. Dr. h. c. Andrej M. Kielbassa

Academic Editor

Kind regards,

Andrej M Kielbassa

Academic Editor

PLOS ONE

Reviewers' comments:

Reviewer's Responses to Questions

**Comments to the Author**

1. If the authors have adequately addressed your comments raised in a previous round of review and you feel that this manuscript is now acceptable for publication, you may indicate that here to bypass the “Comments to the Author” section, enter your conflict of interest statement in the “Confidential to Editor” section, and submit your "Accept" recommendation.

Reviewer #1: All comments have been addressed

Reviewer #2: All comments have been addressed

Reviewer #3: All comments have been addressed

2. Is the manuscript technically sound, and do the data support the conclusions?

Reviewer #1: Yes

Reviewer #2: Yes

Reviewer #3: (No Response)

3. Has the statistical analysis been performed appropriately and rigorously? 

Reviewer #1: Yes

Reviewer #2: Yes

Reviewer #3: (No Response)

4. Have the authors made all data underlying the findings in their manuscript fully available?

Reviewer #1: Yes

Reviewer #2: Yes

Reviewer #3: (No Response)

5. Is the manuscript presented in an intelligible fashion and written in standard English?

Reviewer #1: Yes

Reviewer #2: Yes

Reviewer #3: (No Response)

6. Review Comments to the Author

Reviewer #1: This revised and re-submitted draft is considered ready for external review.

Reviewer #2: (No Response)

Reviewer #3: (No Response)

7. PLOS authors have the option to publish the peer review history of their article (what does this mean?). If published, this will include your full peer review and any attached files.

Reviewer #1: No

Reviewer #2: No

Reviewer #3: No

---

## [Editor Report · Acceptance letter]

26 Jul 2022

PONE-D-21-39587R2 

Is examining children and adolescents with autism spectrum disorders a challenge? - Measurement of Stress Appraisal (SAM) in German dentists with key expertise in paediatric dentistry 

Dear Dr. Reis:

I'm pleased to inform you that your manuscript has been deemed suitable for publication in PLOS ONE. Congratulations! Your manuscript is now with our production department. 

Kind regards, 

on behalf of

Prof. Dr. med. dent. Dr. h. c. Andrej M Kielbassa 

Academic Editor

PLOS ONE